# Drought-Stress-Related Reprogramming of Gene Expression in Barley Involves Differential Histone Modifications at ABA-Related Genes

**DOI:** 10.3390/ijms241512065

**Published:** 2023-07-27

**Authors:** Charlotte Ost, Hieu Xuan Cao, Thuy Linh Nguyen, Axel Himmelbach, Martin Mascher, Nils Stein, Klaus Humbeck

**Affiliations:** 1Institute of Biology, Martin Luther University Halle-Wittenberg, Weinbergweg 10, 06120 Halle, Germany; 2Forest Genetics and Forest Tree Breeding, Georg-August University of Göttingen, 37077 Göttingen, Germany; 3Leibniz Institute of Plant Genetics and Crop Plant Research (IPK), Corrensstraße 3, Gatersleben, 06466 Seeland, Germany; 4German Centre for Integrative Biodiversity Research (iDiv) Halle-Jena-Leipzig, Puschstraße 4, 04103 Leipzig, Germany; 5Center of Integrated Breeding Research (CiBreed), Georg-August University of Göttingen, 37073 Göttingen, Germany

**Keywords:** ABA, drought stress, histone modifications, epigenome, transcriptome, *Hordeum vulgare*

## Abstract

Plants respond to drought by the major reprogramming of gene expression, enabling the plant to survive this threatening environmental condition. The phytohormone abscisic acid (ABA) serves as a crucial upstream signal, inducing this multifaceted process. This report investigated the drought response in barley plants (*Hordeum vulgare*, cv. Morex) at both the epigenome and transcriptome levels. After a ten-day drought period, during which the soil water content was reduced by about 35%, the relative chlorophyll content, as well as the photosystem II efficiency of the barley leaves, decreased by about 10%. Furthermore, drought-related genes such as *HvS40* and *HvA1* were already induced compared to the well-watered controls. Global ChIP-Seq analysis was performed to identify genes in which histones H3 were modified with euchromatic K4 trimethylation or K9 acetylation during drought. By applying stringent exclusion criteria, 129 genes loaded with H3K4me3 and 2008 genes loaded with H3K9ac in response to drought were identified, indicating that H3K9 acetylation reacts to drought more sensitively than H3K4 trimethylation. A comparison with differentially expressed genes enabled the identification of specific genes loaded with the euchromatic marks and induced in response to drought treatment. The results revealed that a major proportion of these genes are involved in ABA signaling and related pathways. Intriguingly, two members of the protein phosphatase 2C family (PP2Cs), which play a crucial role in the central regulatory machinery of ABA signaling, were also identified through this approach.

## 1. Introduction

The sessile lifestyle of plants burdens numerous challenges, including the accessibility of water. Water shortage leads to drought stress, affecting the plant on a morphological, physiological and molecular level during every stage of development. Plants exhibit a multitude of responses to upcoming drought conditions, including maintaining stomatal conductance by osmotic adjustment due to the accumulation of osmolytes like proline, sugars and phenols [1,2] or altering of the root system [3,4,5]. One of the first responses to the perception of water shortage is stomatal closing, thereby minimizing the loss of water through stomata but also reducing photosynthetic action through diminished CO_2_ uptake. The hormone abscisic acid (ABA) plays a major role in regulating the drought stress response in plants. Decreasing water availability leads to an induction of expression of ABA biosynthesis-related genes [6,7], e.g., the enzymes NCED3 (NINE-CIS-EPOXYCAROTENOID DIOXYGENASE 3), ZEP (ZEAXANTHIN EPOXIDASE), AAO3 (ALDEHYDE OXIDASE) and MCSU (MOLYBDENUM COFACTOR SULFURASE). The newly synthesized ABA then interacts with proteins of the PYR/PYL/RCAR (PYRABACTIN RESISTANCE/PYRABACTIN RESISTANCE-LIKE/REGULATORY COMPONENTS OF THE ABSCISIC ACID RECEPTOR) family, which function as ABA receptors and thereby inhibit clades A PP2Cs (protein phosphatases 2Cs), like ABI1 (abscisic acid insensitive 1), ABI2, HAB1 (hypersensitive to ABA1) and HAB2 [8,9,10,11]. The inhibition of the repressor PP2Cs leads to an activation of SnRK2 (sucrose nonfermenting 1-related kinase 2-type protein kinases), including the OST1 (open stomata 1)/SnRK2-6 and the SLAC1 (S-type anion channel), which function in the induction of stomatal closure and other stress responses [12,13,14,15,16].

Recent studies revealed that responses of plants to environmental stress involve higher-order epigenetic regulatory mechanisms, such as differential histone modifications and DNA methylation, affecting chromatin structure (reviewed in [17]). Especially in response to drought, fast and dynamic changes in histone modifications, as well as more stable and progressive changes in the histone context, were reported [18,19,20,21]. However, the exact mechanistic link between epigenetic modifications and transcriptional regulation is not yet fully understood [18,22]. Most work concerning stress-related histone modifications was undertaken with the model plant *Arabidopsis thaliana*. Using chromatin immunoprecipitation followed by deep sequencing (ChIP-Seq), genome-wide distribution patterns of mono-, di- and trimethylation of H3K4 in *A. thaliana* under drought stress were conducted, and it was shown that in contrast to the moderately altered H3K4me1 and H3K9me2 marks, H3K4me3 changed prominently, corresponding with differentially expressed genes [23]. Furthermore, dehydration- and ABA-inducible genes showed a broader distribution of H3K4me3 over the gene body. In another genome-wide study in rice under drought stress, the transcript level of a subset of stress-responsive genes could be positively correlated with the modification level of H3K4me3 [24]. Mild prolonged stress was applied on *Zea mays*, followed by a recovery phase while examining changes in the H3K4me3, H3K9ac and H3K27me3 levels with an additional focus on transcript-level changes. It was shown that between 25 and 30% of genes marked with H3K4me3 or H3K9ac are also upregulated during drought, whereas the H3K27me3 mark showed no correlation [25]. In the grass *Brachypodium distachyon*, combined ChIP-Seq and transcriptome data could identify genes upregulated during PEG-6000-simulated drought stress and associated with an increased H3K9ac level [26].

In this study, the genome-wide responses of the crop plant *H. vulgare* to drought stress at the levels of H3K4 trimethylation and H3K9 acetylation were investigated, showing epigenetic control of ABA-related stress responses.

## 2. Results

### 2.1. Set-up of Drought Stress Experiment

Barley (*H. vulgare* cv. Morex) plants, grown on soil, were exposed to drought by withholding water. This resulted in a continuous decrease in soil water content (Figure 1A). The other plants were kept under control conditions at about 65% soil water content. This mild drought stress caused an early decrease in chlorophyll content and photosystem II (PSII) efficiency in the primary leaf (Figure 1B), indicating premature stress-induced senescence processes with decreasing photosynthesis and chloroplast degradation. Exemplarily, in Figure 1C, primary leaves of drought-stressed and control plants at different time points are presented. The drought-stressed leaves clearly show a much earlier loss of chlorophyll content compared to the control leaves. Developmental senescence of primary leaves in the control starts later, around day 37. As indicated in Figure 1B, primary leaves for gene expression analyses were harvested from both drought-stressed and control plants before drought treatment (M0) and at five stages of drought-stress treatment (D1-D5). In control plants, leaves were additionally harvested at two further stages (S2 and S3). The developmental and drought stress stages were defined based on the relative chlorophyll content. The maximum measured value of the chlorophyll content was set as 100%, representing the mature stage in which the primary leaf is fully developed (M0). Leaves of stage D1/S1 have a relative chlorophyll content of 95%, D2/S2 90%, D3/S3 75%, D4/S4 50% and D5/S5 25%. In control plants, stage S1 is on the same day as D4 in drought-stressed samples.

### 2.2. Expression of Stress- and Senescence-Related Marker Genes

Drought stress results in a major reprogramming of gene expression [27]. To map this process, the expression of known marker genes of drought stress and leaf senescence was analyzed (Figure 2). These marker genes include the senescence-associated gene *HvS40* [28,29,30]; *HvA1*, encoding a dehydrin of the LEA protein family [31,32]; *P5CS2*, encoding a Delta-1-pyrroline-5-carboxylate synthase 2 enzyme involved in proline biosynthesis [33,34,35]; and *Hsp17*, encoding a stress-related heat shock protein [36]. *HvS40*, *HvA1*, *P5CS2* and *Hsp17* are upregulated in response to drought treatment and, to some extent, during developmental senescence.

### 2.3. Drought Stress Specifically Alters Genome-Wide Loading with Euchromatic Marks H3K4me3 and H3K9ac

Primary leaves were harvested at three different stages (M0 and M2 as controls and D2 as the drought-stressed sample). To detect early responses to drought, the D2 stage was chosen, which represents a rather early phase of drought stress, with only a 10% loss of chlorophyll and less than a 5% decrease in photosynthetic performance (Figure 1B). Chromatin immunoprecipitation (ChIP) followed by Illumina deep sequencing for each stage was performed with samples from three independent experiments to detect the genome-wide loading of histone H3 with the euchromatic marks K4me3 and K9ac in M0, M2 and D2. At an average count of 18.9 million reads per sample, 93% of the reads could be mapped to the barley genome MorexV2 [37,38]. The mapping statistics for the input (untreated chromatin) and the IP samples (antibody-treated samples) are listed in Appendix A. To detect enriched regions of H3K4me3 and H3K9ac, pooled peak calling with all three IP samples and their corresponding input samples was performed with MACS2 (see Material and Methods for details), yielding lists of peaks for each time point and histone modification. The detected peaks were intersected with the annotated gene list of the barley genome MorexV2, resulting in ~7000–28,000 genes associated with a mark for each sample (Figure 3A, genes are listed in Appendix A). The pure number of detected genes with H3K4me3 loading is much higher than with H3K9ac loading, and most of the detected genes (15,302) share both marks. Peak distribution covering all seven chromosomes (1H–7H) is shown in Figure 3B. The euchromatic marks H3K4me3 and H3K9ac detected in this analysis agglomerate at chromosome end regions and not at the centromere, showing the same distribution as transcriptionally active genes [37]. In addition, we analyzed the heterochromatic mark H3K9me2, which shows the opposite distribution and is spread over centromeric, inactive regions and is missing at the active chromosome ends.

To identify only those genes with prominent H3K4me3 and H3K9ac loading at the promoter and gene body, a stringent threshold for peak detection was applied. MACS2-identified peaks with a fold enrichment ≥10-fold and a *q*-value < 0.05 were selected. Figure 3C shows Venn diagrams comparing these gene lists (Appendix A) identified from leaves under control conditions (M0 and M2) and under drought stress conditions (D2). For both marks, around 50% of the genes loaded with the euchromatic marks (H3K4me3 and H3K9ac) are common in all three conditions M0, M2 and D2 (15,089 genes with H3K4me3 and 6174 genes with H3K9ac). However, there are genes specifically loaded only in one or two conditions. Interestingly, the majority of genes loaded with H3K4me3 and H3K9ac specifically in only one condition are found in M0, meaning that these marks are erased during a further 8 days of development (from 11 to 19/21 days after sowing), irrespective of control or drought conditions. Comparing the two different euchromatic marks, there are differences in changes during development and drought stress. While 5993 genes are labeled with H3K4me3 in both developmental stages, M0 and M2, only 751 genes retain H3K9ac marks during development from M0 to M2. This could indicate that H3K9ac labeling is more flexible than H3K4me3 labeling during development, respectively. Interestingly, under drought stress, 2008 genes were specifically labeled with H3K9ac compared to only 129 genes labeled specifically with H3K4me3, indicating again higher flexibility of H3K9 acetylation than H3K4 trimethylation. 

Euchromatic marks like K4 trimethylation and K9 acetylation are known to be preferentially positioned at histones 3, associated with DNA around transcriptional start sites (TSSs) and reach into the gene body [39]. This distinct spatial distribution of transcriptionally active marks is also seen for the identified genes associated with H3K4me3 or H3K9ac (Figure 3D). Using deepTools, mean scores calculated from the signal tracks of the histone enrichment were plotted against the genes associated with a peak. In all three conditions (M0, M2 and D2), the highest enrichment of H3K4me3 and H3K9ac is around TSS and the adjoining gene bodies. While H3K4me3 shows broader labeling with high loading over the whole gene body with a sharp drop at the transcription end site (TES), H3K9ac is preferentially positioned around TSSs. This was also described in a previous study in barley, where strong enrichment in the ChIP-Seq peaks around the TSSs for H3K4me3 and H3K56ac could be observed [40].

### 2.4. Functional Analysis of Genes Loaded with H3K9ac or H3K4me3 in Response to Drought 

Comparing the genes associated with the euchromatic marks at the different time points revealed that in response to early drought, 129 genes were loaded with H3K4me3 and 2008 genes with H3K9ac (Figure 3C). Using the GO enrichment analysis platform TRAPID [41], these genes (listed in Appendix A) were functionally clustered, and enrichment analyses were performed. The program uses the Benjamini and Hochberg method for correcting, and the maximum q-value was set to 0.05. While, with these settings, no significantly enriched GO terms could be detected for the low number of H3K4me3-marked genes, this analysis allowed the identification of specific functional classes of genes marked with H3K9ac in response to drought stress. 

Figure 4A shows the top ten enriched GO terms for biological process and molecular function for the 2008 genes associated exclusively with H3K9ac in D2 (see full list in Appendix A). These genes have functions in several metabolic processes, involving small molecule metabolism, organophosphate metabolism, lipid metabolism and endopeptidase function. In addition, genes were found to be involved in plastid organization, embryo and epidermis development and protein binding. The most striking set of genes with the highest q-value is involved in response to abiotic stimulus. This set of 193 genes, specifically loaded in response to drought with the euchromatic mark H3K9ac, is functionally connected to different abiotic stress responses, including drought/osmotic stress, light stimulus, salt stress, temperature stress, cold stress and external stimulus (Figure 4A). Appendix A shows an increase in H3K9ac loading for 10 well-known stress-related genes. For instance, a 9-cis-epoxycartenoid dioxygenase (a central enzyme of ABA biosynthesis) and two ethylene-responsive transcription factors, involved in stress response, all demonstrate a clear increase in acetylation at K9 during drought. To validate the deep sequencing results, loading with H3K4me3 and H3K9ac at five genes from that list was additionally validated via qRT-PCR (Figure 4B). These genes are a protein phosphatase 2C (*PP2C*), a bidirectional sugar transporter SWEET (*SWEET*), a bZIP transcription factor (*bZIP TF*), an ABA receptor and a heat shock transcription factor (*HS TF*). This analysis revealed a significant enhancement in both marks in drought-stressed samples, with PP2C showing the strongest enrichment with both euchromatic marks. 

### 2.5. Identification of Genes Specifically Labeled with H3K4me3 and H3K9ac and being Upregulated during Drought Stress

Via genome-wide ChIP-Seq, genes that were loaded with the euchromatic marks H3K4me3 and H3K9ac during development (M2vsM0) and in the early phase of drought stress (D2 vs. M2) were identified. Additionally, using a genome-wide RNA-Seq approach, parallel changes in the transcriptional activity of barley genes during development and drought stress were detected. Lists with differentially expressed genes (DEGs) under these conditions are provided in the Appendix A. Using stringent conditions (an adjusted *q*-value < 0.05; a log fold change ≥|1|), 220 differentially expressed genes in response to early drought stress (D2 vs. M2) could be identified, with 103 being downregulated and 117 being upregulated (Figure 5A). RNA-Seq data were validated for selected genes via qRT-PCR (Figure 5B). To gain insight in the biological functions of the upregulated genes, gene ontology enrichment analysis was performed. The main enriched GO terms are cell-wall-related (plant-type secondary cell wall biogenesis, a lignin catabolic process), stress-related (response to osmotic stress and response to water deprivation), the phenylpropanoid metabolic process, response to abiotic stimulus and oxidation-reduction processes. 

Using stringent conditions, a comparison of ChIP-Seq data with transcriptome data revealed that from the 117 genes upregulated at the early phase of mild drought stress (D2), 10 genes could be identified via ChIP-Seq to be loaded with H3K4me3 and 12 genes with H3K9ac (listed in Figure 5C). Many of these genes are involved in abiotic stress responses, e.g., a Myb-related transcription factor, cytochrome P450 and the bidirectional sugar transporter SWEET. In Figure 5D, the loading of three genes with H3K4me3 and three genes with H3K9ac is illustrated as examples. Interestingly, two members of the PP2C family were shown to be loaded with H3K9ac and upregulated early during drought stress. 

These protein phosphatases 2C (PP2Cs), which play a central role in the ABA signaling module, were analyzed in more detail. They function as a negative regulatory switch at the center of the ABA signaling network, tuning the ABA response and integrating it with other developmental and stress-related pathways [42]. In Arabidopsis, 76 *PP2C* genes have been identified and grouped into ten clades (A–J), with six of nine *PP2Cs* belonging to clade A being involved in ABA signaling [43]. Previous studies in Arabidopsis have shown that the expression of these genes is regulated by epigenetic mechanisms, altering the chromatin state [44,45]. In *H. vulgare* cv. Morex, 85 *PP2C* genes were recently identified [46]. In this study, it was shown, that a total of 26 *H. vulgare PP2Cs*, including 6 putative and 6 family protein *PP2Cs*, are already loaded with the euchromatic acetylation mark at K9 during early drought stress (see Appendix A). Comparing the expression levels of the *PP2Cs* with their corresponding K9ac signal as a heatmap, it was observed that a higher enriched signal correlates with an enhanced expression (Figure 6A). Blast analysis in combination with the recent PP2C phylogenetic studies in hulless barley [47] revealed that six of the top seven K9ac-enriched *PP2Cs* belong to clade A. Validation via qRT-PCR of three selected *PP2Cs* confirmed these results (Figure 6B). Subsequent expression analyses for the more severe drought stress stages D3 and D4 showed an increasing relative transcript level for all three *PP2Cs*. 

## 3. Discussion

The experimental set-up of this study allowed for a slow decrease in the relative soil water content, achieved by withholding water at day 11, to simulate more natural drought stress. While a majority of studies concentrated on proceeded drought stress in barley [48,49,50,51], the main focus of this research was on the epigenetic and transcriptomic responses in the early phase after the onset of drought. Therefore, sampling time points were chosen at a very early stage, when only 10–15% of the chlorophyll and only 5–10% of the photosynthetic performance were lost. Transcriptomic analysis via qRT-PCR revealed that at this time point, marker genes of drought stress were already induced. 

### 3.1. Early Response to Drought Stress Includes Global Reorientation of Histone Modifications H3K9ac and H3K4me3 

With the aid of the ChIP method followed by deep sequencing, drought-responsive changes in the global distribution of the histone modifications H3K4me3 and H3K9ac were detected. Both euchromatic marks, in contrast to the heterochromatic mark H3K9me2, were associated with areas of active gene transcription, which in barley are located toward the telomeres of all seven chromosomes. The spatial distribution of these marks at open and transcriptionally active chromatin has been reported before for different plant species [52]. Generally, H3K4me3 was found at twice as many genes as H3K9ac, which was also reported in previous studies on *A. thaliana* during developmental leaf senescence and in *Paulownia fortunei* [53,54]. Additionally, the majority of genes marked with H3K9ac (98, 4%) were simultaneously marked with H3Kme3. However, the loading with H3K9ac was much more flexible and, in contrast to the loading with H3K4me3, sensitively responded to the development and onset of stress. While the comparison of the three samples (M0, M2 and D2) revealed that there is no major difference in the distribution of H3K4me3 in response to the onset of drought, the distribution of H3K9ac was altered. This indicates that the trimethylation of H3K4 is a widespread epigenetic mark in barley, but the acetylation of H3K9, at least under the conditions we investigated, seems to be a more dynamic histone modification that sensitively reacts to environmental cues. It has been shown that the acetylation of K9 at the drought-inducible gene related to AP2.4 (RAP2.4) occurred strongly after 1 h of drought treatment, whereas the trimethylation of H3 accumulated gradually [20]. Suggestions were made that a fast response to environmental stress via acetylation and a longer-term response of methylation regarding flowering or stress memory could be beneficial [55]. Interestingly, real-time qPCR analyses with drought-stressed rice seedlings showed an elevated expression of four histone acetyltransferases (*HATs*; *OsHAC703*, *OsHAG703*, *OsHAF701*, and *OsHAM701*), and supporting Western blot analysis consequently revealed an enrichment of acetylation on H3K9, K18 and K27, as well as H4K5, in parallel to the increased *OsHAT* expression [56].

The genes that specifically gained euchromatic H3K9ac marks in early drought stress comprise, among the genes involved in several metabolic processes, many stress-related genes. These include genes involved in response to different stimuli, such as cold, light, salt and drought, with several of them being involved in ABA-related stress responses (see Appendix A). This indicates that the early response to drought in the model crop plant *H. vulgare* entails higher-order regulation of the expression of genes involved in abiotic stress responses through differential histone modifications. These findings are supported by the results of genome-wide ChIP sequencing in drought-stressed *Brachypodium distachyon*, where it was shown that the level of H3K9ac is increased at drought-responsive genes [26]. For barley, the levels of H3K4me3 are increased at the heat shock protein 17 (*HSP17*), whereas H3K9me2 is reduced [57]. Recently, overexpression analyses of the WHIRLY1 protein in drought-stressed barley leading to a decrease in H3K9ac and H3K4me3 levels at the ABA-related genes *HvNCED1* and *HvS40* revealed its possible regulatory role toward drought stress responses by interacting with epigenetic regulators [58]. There are further indications that the regulation of drought-stress-related genes is under epigenetic control. Loss-of-function studies of the *Arabidopsis* TRITHORAX-like factor ATX1, which trimethylates H3K4, leads to decreased levels of H3K4me3 at NCED3, a key enzyme in ABA biosynthesis, in drought-stressed *Arabidopsis* plants, resulting in reduced ABA concentration [59]. Furthermore, it has been shown that during dehydration, *NCED3* showed increased enrichment of nucleosomal H3K4me3 [60]. In two studies, enriched H3K4me3 and H3K9ac levels were observed at the drought-inducible genes *RD20* and *RD29A*, where the enrichment levels correlated with the intensity of the stress (moderate vs. severe) [18]. Interestingly, during rehydration, the H3K9ac mark diminished fast and robustly, while H3K4me3 decreased progressively [19]. 

### 3.2. Genes Loaded with Euchromatic H3K9ac and Induced at Early Drought Stress 

ChIP-Seq analyses revealed that specific sets of barley genes were loaded with the euchromatic mark H3K9ac at the early stages of drought stress. It has been previously reported that severe drought stress causes stronger enrichment/changes in the histone marks than mild drought stress [20,61]. To avoid the secondary effects of prolonged drought treatment, this work aimed to detect early responses to drought. An interesting set of genes was identified, loaded with euchromatic histone modifications in early drought conditions, corresponding to their respective upregulation. To identify these genes, RNA-Seq was performed with the same samples used for ChIP-Seq, comparing the transcriptome of drought-stressed plants with that of control plants. As expected under these early and mild conditions, RNA-Seq revealed a relatively small number of DEGs. The intention of this work was to identify particularly those genes that respond quickly to upcoming drought stress and may have basic functions in establishing drought resilience in barley. The intersection of ChIP- and RNA-Seq data revealed that about 11% of the early upregulated genes were loaded with H3K9ac in response to the onset of drought. Similar results have been reported for drought-stressed maize, wherein 25–30% of the genes are marked with increased H3K4me3 or H3K9ac, corresponding to higher expression levels [25]. ChIP-Seq and qRT-PCR analysis of PEG-6000-treated *B. distachyon* revealed 40 genes with increased H3K9ac levels, of which 23 showed elevated transcription levels [26]. Similar results were also observed in rice [24]. In senescing *Arabidopsis* leaves, 22% of the genes that are upregulated during senescence also gain the trimethylation of K4, and only 2% of these genes show elevated H3K9ac marks [53]. 

### 3.3. PP2Cs and Other ABA-Related Genes Are Loaded with H3K9ac in Response to Drought

Interestingly, about half of the genes loaded with euchromatic marks and being upregulated during the early stress response encode proteins involved in stress responses and ABA signaling. Among them are two Myb or Myb-related transcription factors, cytochrome P450, the sugar transporter SWEET and the PP2Cs involved in drought stress signaling [62,63,64]. Interestingly, two members of the PP2C family, involved in the central ABA signaling module [65], were also found to be upregulated and loaded with H3K9ac. This indicates an epigenetic control level as part of drought-stress-induced reprogramming of gene expression through the establishment of euchromatic marks. 

The regulation of *PP2C* genes was further investigated, including the later stages of drought stress. Among the 26 annotated *PP2C* genes associated with H3K9ac in drought, seven were upregulated, with two of them showing significant upregulation at early phases, indicating that several *PP2Cs*, being central regulators in ABA signaling after the onset of drought, are loaded with euchromatic marks and thus upregulated. Moreover, a comparison of the TPM expression values and the signal enrichment of H3K9ac revealed a slight correlation between the expression and the acetylation of several *PP2C* genes. *PP2Cs* with higher TPM values in D2 also depict a stronger acetylation in D2, emphasizing the activating role of K9ac in gene expression [66,67,68]. Notably, six of the seven top K9ac-enriched *PP2Cs* belong to clade A. Recent phylogenetic and transcriptomic studies in Tibetan hulless barley revealed that most of the upregulated *PP2Cs* during dehydration stress belong to clades A or F [47]. Similar results were seen in maize, where a majority of the tested clade A *ZmPP2Cs* were induced under drought or ABA treatment [69,70]. The upregulation of the PP2C-As *ABI1* and *ABI2* during stress conditions (salt, drought and osmotic stress) and the downregulation of the ABA receptors *RCAR3* and *RCAR10* in *Arabidopsis* suggest that higher PP2C levels desensitize the plant to high ABA levels in a negative feedback loop mechanism [71,72,73]. Subsequent experiments confirmed that an increased PP2C:PYR/PYL ratio is essential for the activation of the downstream ABA signaling cascade [74]. Similar results were also demonstrated in barley [75]. 

RNA-Seq analysis of the protein phosphatase 2C (*5HG0392330*) showed a nonsignificant upregulation (logFC = 4.9, padj = 0.07) in drought, while qRT-PCR analysis confirmed upregulation in D2 and an increasing relative gene expression during ongoing drought stress (stage D3 and D4). Similar to the other *PP2Cs* that were upregulated during stress and showed enhanced acetylation in D2, this *PP2C* gene displayed a strong enrichment of H3K9ac but also H3K4me3 at the promotor region and the gene body in D2. BLAST analysis revealed a homology to the rice PP2C *OsPP108/OsPP2C68*, which is highly upregulated under ABA, salt and drought stress, and overexpression of this *PP2C* in rice leads to ABA insensitivity [76].

## 4. Material and Methods

### 4.1. Plant Material and Growth under Drought Stress Conditions

Barley (*H. vulgare* cv. Morex) seeds, obtained from IPK Gatersleben (OT Gatersleben, Seeland, Germany), were incubated for 72 h at 4 °C in darkness, followed by 24 h at room temperature on wet tissue paper. Germinated seedlings were sown as described by Temel et al. [57] with slight modifications. Fifty Mitscherlich pots, each containing 12 seedlings and 1.5 kg ED73 soil (Einheitserdewerke Werkverband e.V., Sinntal-Altengronau, Germany), were placed in a greenhouse cabinet and grown under long-day conditions (16 h light, 23° C/8 h dark, 18 °C, light intensity 100 µmol m^−2^ s^−1^ and 45–50% relative humidity). On the twelfth day after sowing (das), the water for half of the pots was withheld to induce drought stress, while the other half served as the control group and was watered. Every two days, the pots were weighed to calculate the soil water content, and physiological parameters were measured. The experiment was performed eight times. For RNA expression analysis and RNA-sequencing, primary leaves were harvested, frozen in liquid nitrogen and stored at −80 °C. For the chromatin immunoprecipitation (ChIP), 1 g of primary leaves was harvested and fixed with formaldehyde before being frozen in liquid nitrogen and stored at −80 °C. 

### 4.2. Physiological Measurements

The relative chlorophyll content and the PSII efficiency were measured every second day between 7 and 47 das. The relative chlorophyll content was measured using the SPAD (Soil Plant Analysis Development) tool from Minolta (Konica Minolta Sensing Europe B.V., Munich, Germany). The relative chlorophyll content of 20 primary leaves from 20 plants was measured at the top, middle and bottom of the leave, and the mean value was calculated. Each data point represents the mean of at least five biological replicates. The chlorophyll fluorescence is the difference between the maximum (F_m_) and minimum (F_0_) fluorescence and is called variable fluorescence (F_v_). The PSII efficiency was calculated using the following formula: (F_m_ − F_o_)/F_m_ = F_v_/F_m._ Leaves were dark adapted for 10 min, and PSII efficiency was measured using the MINI-PAM fluorometer (Walz GmbH, Effeltrich, Germany).

### 4.3. Gene Expression Analysis Using Quantitative Real-Time PCR (qRT-PCR)

Total RNA from at least four independent biological replicates was isolated with homemade TRIzol reagent following the protocol, as described by Chomczynski and Mackey [77]. In general, all DNA and RNA concentrations were measured using a nanospectrophotometer (NanoPhotometer^®^ NP80, Implen, Munich, Germany). Samples for sequencing were additionally measured using a Bioanalyzer2100 system (Agilent Technologies, Santa Clara, CA, USA) using the RNA 6000 pico kit and the HS DNA kit (Agilent Technologies, Santa Clara, CA, USA) and with a Qubit 4 Fluorometer (Thermo Fischer Scientific, Waltham, MA, USA). With the RevertAid^TM^ H Minus First Strand cDNA Synthesis Kit (Thermo Fisher Scientific, Waltham, MA, USA), 1 µg of total RNA was transcribed into cDNA following the manufacturer’s instructions. For qRT-PCR, 2 µL of cDNA template was mixed with the KAPA SYBR fast qPCR Mastermix (KAPA Biosystems, Inc., Wilmington, MA, USA) and the gene-specific primers (Appendix A). To exclude the amplification of unspecific products, a no RT (reverse transcriptase) control was additionally run. Quantitative RT-PCR was performed using the CFX Connect Real-Time PCR Detection System from Bio Rad (Bio-Rad Laboratories Inc., Göttingen, Germany). For the determination of the relative gene expression, the REST-384 ©2006 software v2 [78] was used with genes HvActin, HvPP2A and HvGCN2 (constant expression under control and drought conditions) as reference genes for normalization.

### 4.4. Chromatin Immunoprecipitation (ChIP) Followed by Sequencing

The ChIP was performed as described in Ay et al. [79] with some minor modifications. Samples from three different stages were taken: M0 and M2 as control samples, and the corresponding drought stress sample D2. The isolated chromatin was sheared (average fragment size: 300 bp) using the Covaris M220 Focused Ultrasonicator (Covaris LLC, Woburn, MA, USA) with the following settings: duty factor, 10%; peak incident power, 50; cycles/burst, 200; duration, 300 sec; and temperature at 6 °C. Two antibodies against euchromatic H3K4me3 (ab8580) and H3K9ac (ab10812) and one antibody against heterochromatic H3K9me2 (ab1220), all obtained from Abcam (Cambridge, UK), were used for the overnight immunoprecipitation step at 4 °C. In total, ChIP-DNA from three independent replicates was isolated and used for library preparation and sequencing. The libraries were prepared using the Illumina TruSeq ChIP Library Preparation Kit and sequenced (paired-end, 2 × 101 cycles) on the Illumina HiSeq2500 device (both Illumina Inc., San Diego, CA, USA). The analysis of the sequenced data was accomplished with a specific workflow. In brief, after quality control using FASTQC [80], the trimmed reads were aligned to the second version of the reference genome sequence assembly of barley cv. Morex [37,38] with BWA-MEM [81,82] with default parameters. The software MACS2 (version 2.1.1) [83] was used to quantify regions that were enriched in reads (peaks). The peak detection was carried out with all three IP samples from every biological replicate and their corresponding input samples in a pooled peak calling. Resulting peaks with a fold enrichment ≥10 were intersected with the annotated gene list of Morex V2. Signal tracks of the histone modifications enrichment levels over the whole genome were built using the *bdgcmp* command from MACS2, converted into bigWig files on the Galaxy platform [84] with the UCSC BedGraph-to-bigWig converter [85] and could then be visualized in the integrative genome browser (IGV) [86] together with the called peaks. To plot the peak distribution over the signal track profiles of the genes, the software deepTools (version 3.1.3) [87] was used. To compare the list of genes with each other, a web tool creating Venn diagrams was used (https://bioinformatics.psb.ugent.be/webtools/Venn/, accessed on 1 March 2018). For calculating the differences in the signal tracks between M2 and D2, deepTools’ multiBigwigSummary and the Diffbind package [88] were used.

To validate the detected peaks, quantitative real-time PCR of the ChIPed-DNA was performed with five different gene-specific primer sets (Appendix A). To normalize the ChIP-qPCR data, the percent input method was used, in which the enrichment of the IP samples is determined as the % of the input [89].

### 4.5. RNA-Seq Analysis

Total RNA from four independent replicates were isolated with TRIzol reagent followed by a purification step using the RNeasy^®^ Plant Mini Kit (Qiagen, Hilden, Germany). Library preparation was undertaken with the Illumina TruSeq RNA Library Preparation Kit (Illumina Inc., San Diego, CA, USA) according to the manufacturer’s instructions. The samples were sequenced (paired-end, 2 × 101 cycles) with the Illumina HiSeq2500 instrument (Illumina Inc., San Diego, CA, USA). Like the ChIP-Seq data, the analysis of the RNA-Seq data was accomplished with a workflow. In brief, as a first step after read quality control, the transcript abundance was quantified by pseudo alignment of the processed reads with the software Kallisto (v0.45.0) [90]. The Kallisto commands were executed with the default parameters, except for bootstrap samples, which were set to 40. The resulting abundance data from the Kallisto pseudo alignment were imported into R (R core team, 2018, version 3.5.1) for statistical analysis [91]. The following steps were executed with the R packages edgeR [92] and limma [93]. The genes that are differentially expressed between control and drought stress samples were identified. Genes were considered differentially expressed when logFC ≥ |1|. The adjusted *p*-value cut-off was set to ˂0.05. The RNA-Seq results were validated using qRT-PCR with four selected primers (Appendix A), as described in Section 4.3. 

### 4.6. GO Term Enrichment Analysis

The GO term enrichment analysis was executed using the TRAPID software (version 2.0) [41], and the maximum *q*-value was set to 0.05. The enrichment bubble plots were plotted using http://www.bioinformatics.com.cn/srplot (accessed on 1 April 2023), an online platform for data analysis and visualization.

## 5. Conclusions

Barley has developed complex mechanisms to adapt to drought stress. One of these mechanisms includes histone modifications, particularly H3K9 acetylation, leading to the upregulation of specific genes (Figure 7). It is shown that during the early phase of drought stress, when chloroplasts remain active, a specific set of genes is loaded with H3K9ac and upregulated. Among these genes are ABA-related genes, such as those encoding *PP2Cs* involved in the central regulatory unit of ABA action. The results suggest that H3K9ac works more flexibly in response to drought compared to K4 trimethylation, indicating a potential role of H3K9 acetylation in the short-stress memory of plants. Furthermore, the findings support the hypothesis that H3K9ac may act as a template for the trimethylation of K4. 

## Figures and Tables

**Figure 1 ijms-24-12065-f001:**
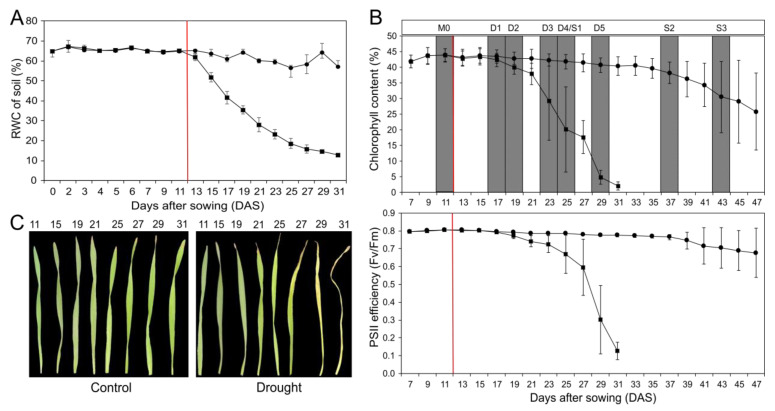
Physiological characterization of barley plants under developmental (●) and drought stress (■) conditions. (**A**) Relative soil water content of control and drought-stressed plants during a period of 31 days. (**B**) Relative chlorophyll content and PSII efficiency (Fv/Fm = maximum quantum yield) of control and drought-stressed plants over a period of 47 days. The red line marks the starting point of water retention. Developmental stages were defined based on the relative chlorophyll content. The maximum measured value of the chlorophyll content was set as 100%, determined as the mature stage in which the primary leaf is fully developed (M0). Every data point corresponds to the mean values of four to eight independent measurements. (**C**) Photographic documentation of primary leaves under drought stress (**right**) and control (**left**) conditions during a period of 31 days. Each data point represents the average value from at least five biological replications, and the error bar shows the mean (±) standard deviation.

**Figure 2 ijms-24-12065-f002:**
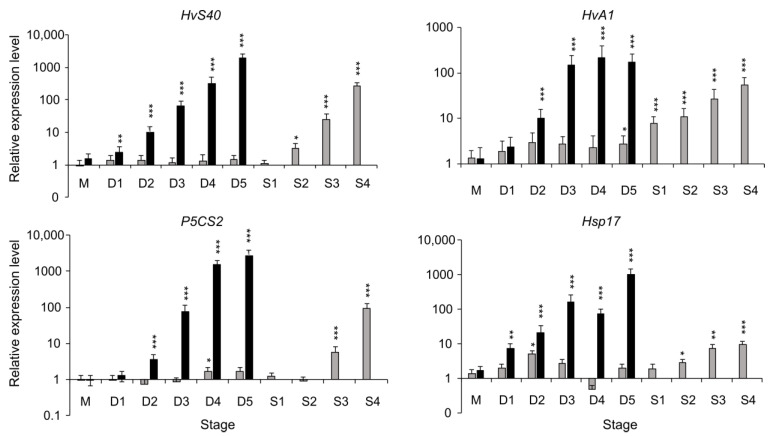
The relative transcript levels of drought-stress-related and senescence-associated genes were measured in mature, nonstressed stage (M) and at different stages during drought stress (D1–D5, represented by (■) and during development under control conditions (■) of mature leaves at the same days when drought-treated plants are at stages D1–D5 and later at different stages of developmental senescence (S1–S4). Each bar in the graph represents the average value from three separate replications, and the error bars show the ±SE. The asterisks above the graph bar indicate statistically significant differences according to Student’s *t*-test (* *p* < 0.05, ** *p* < 0.01 and *** *p* < 0.001). Mean relative expression levels, standard error and *p*-values were calculated using the software REST-384 ©2006 (v2.0, Qiagen GmBH, Hilden, Germany).

**Figure 3 ijms-24-12065-f003:**
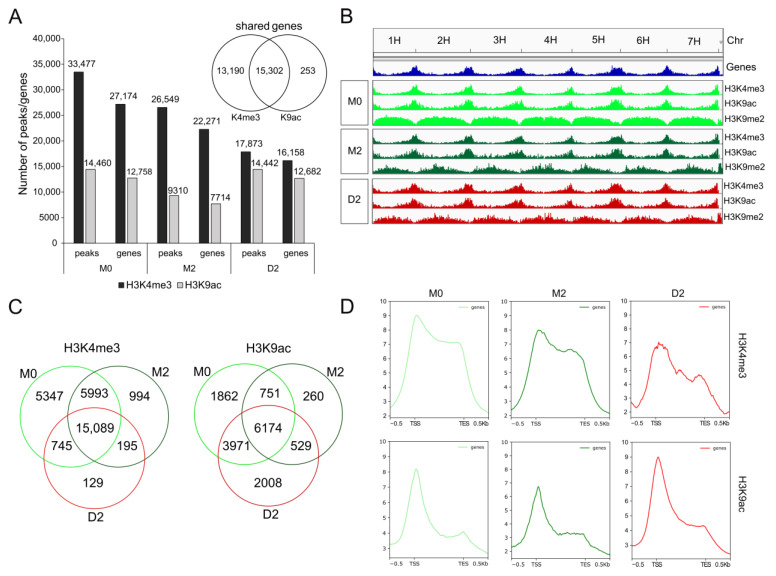
Genome-wide identification of euchromatic marks. (**A**) Number of detected peaks and genes for both modifications and the three time points. (**B**) Visualization of the detected peaks in the integrative genome viewer (IGV). (**C**) Venn diagrams of the comparison of the gene lists loaded with H3K4me3 and H3K9ac between all three time points. (**D**) Distribution of the signal around the TSS of their corresponding genes for H3K4me3 and H3K9ac, where the y-axis depicts the mean normalized log2 ratio between the signal track of the histone enrichment and the genic regions, and the x-axis presents the genomic ranges.

**Figure 4 ijms-24-12065-f004:**
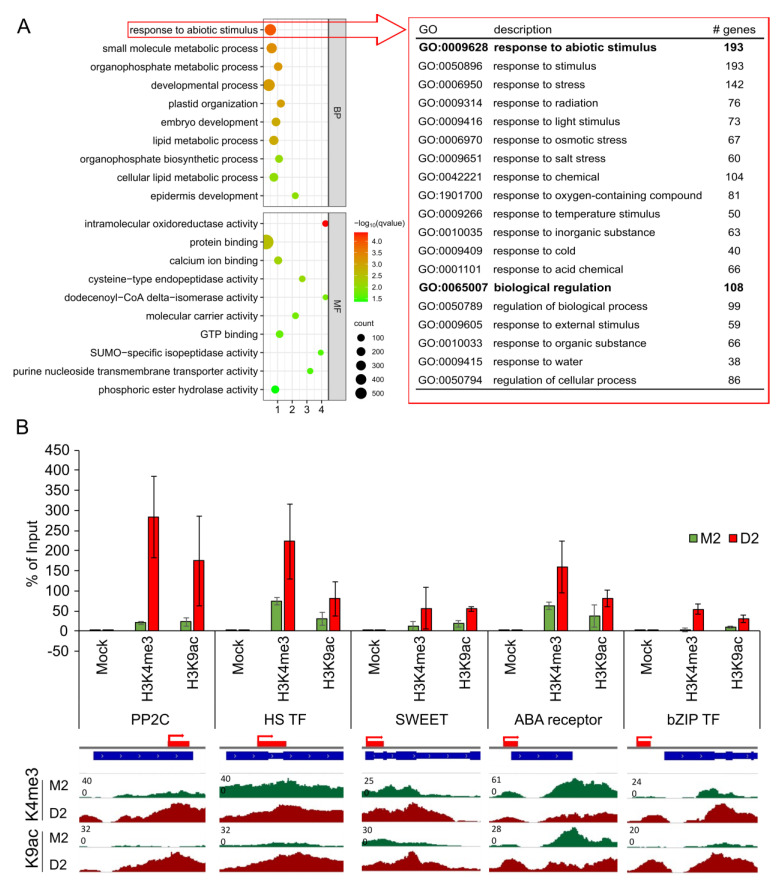
Analysis of genes associated with H3K4me3 or H3K9ac. (**A**) GO enrichment analysis of genes associated with H3K9ac in D2 in comparison to M0 and M2. The top ten GO terms for biological process (BP) and molecular function (MF) were chosen based on the *q*-value. Subset enrichment analysis were conducted for the over-represented term “response to abiotic stimulus”, and the first two terms with corresponding subterms (in red) are shown, including the number (#) of genes in the subset terms. (**B**) ChIP-Seq validation with five selected genes. The percentage of input was calculated for a protein phosphatase 2C (*PP2C*) gene, a heat shock transcription factor (*HS TF*), a bidirectional SWEET transporter (*SWEET*), a gene encoding for an ABA receptor and a bZIP transcription factor (*bZIP TF*) (n = 3). The red bands mark the PCR amplicon. In addition, signal tracks of histone 3 loading with K4me3 or K9ac of promoter and TSS regions of these genes are shown for M2 (green) and D2 (red).

**Figure 5 ijms-24-12065-f005:**
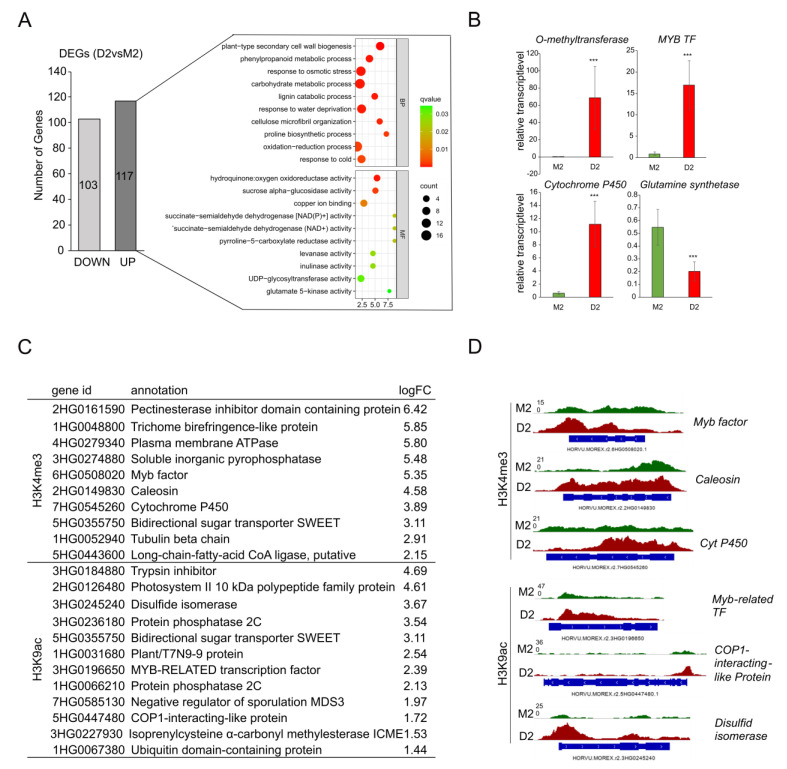
Comparison of transcriptomic changes and histone modifications. (**A**) Number of up- and downregulated genes in drought stress (D2vsM2) and GO enrichment analysis of genes upregulated during drought. (BP) = biological process, (MF) = molecular function. (**B**) Relative transcript level calculated via qRT-PCR of four selected up- and downregulated genes, confirming the results of the RNA sequencing. The asterisks above the graph bar indicate statistically significant differences according to Student’s *t*-test (*** *p* < 0.001). (**C**) Genes associated with a histone mark and being upregulated in D2. (**D**) Signal tracks of selected genes for H3K4me3 and H3K9ac.

**Figure 6 ijms-24-12065-f006:**
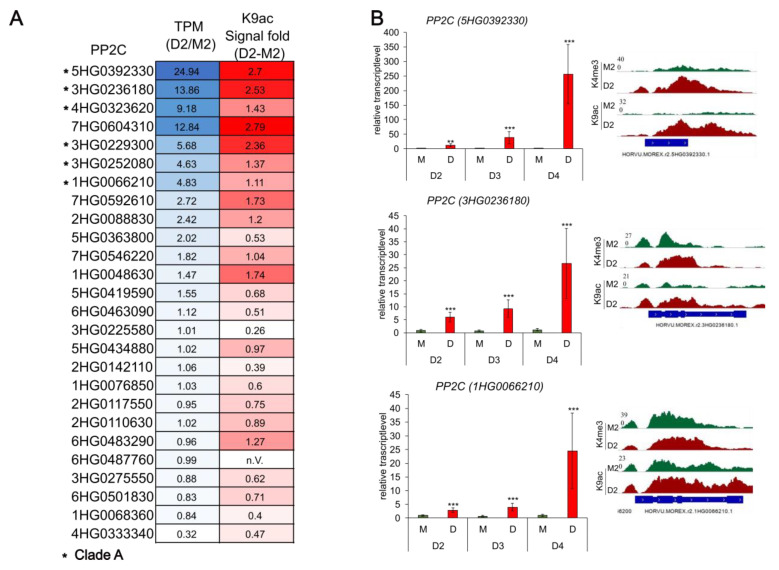
PP2Cs associated with H3K9ac in D2. (**A**) Heatmap of the 26 *PP2Cs* labeled with H3K9ac in D2. Shown are the color-coded values for the TPM (transcript per kilobase million) ratio (D2/M2) in blue and the signal of H3K9ac enrichment (signal fold D2–M2) in red. Darker colors indicate higher values; asterisks mark the *PP2Cs* that belong to clade A. (**B**) Validation of three selected *PP2Cs* via qRT-PCR. The relative transcript levels calculated with the *ct*-values from the qRT-PCR for the stages D2, D3 and D4 are shown, along with the corresponding signal tracks of the histone mark for D2. The asterisks above the graph bar indicate statistically significant differences according to Student’s *t*-test (** *p* < 0.01 and *** *p* < 0.001).

**Figure 7 ijms-24-12065-f007:**
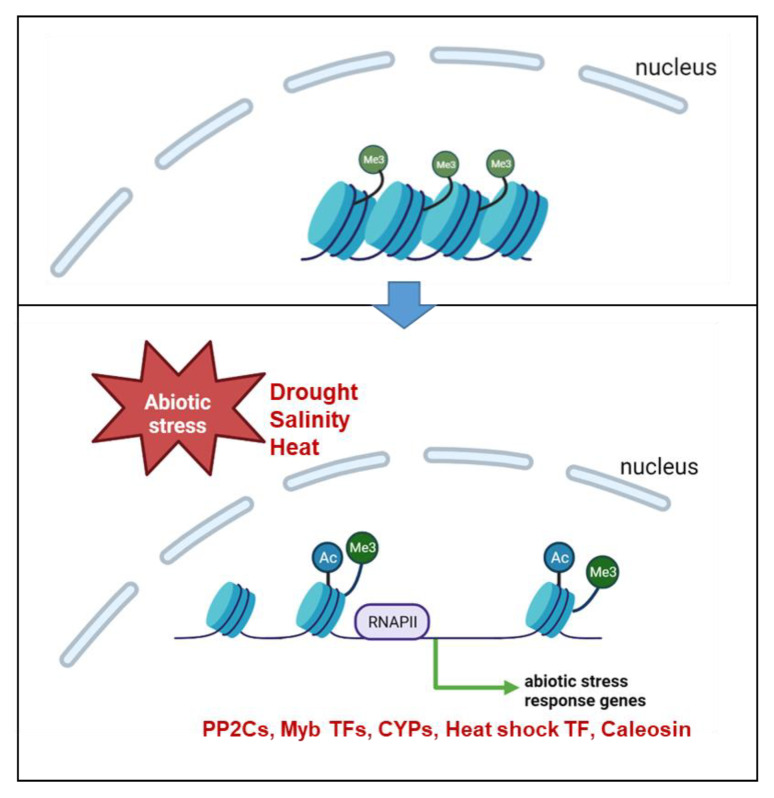
Working model of the changes in histone modifications and chromatin structure during abiotic stress (created using BioRender.com). Shown are the histones modified with H3K4me3 (green circles with “Me3”) and H3K9ac (blue circles with “Ac”) in the nucleus. Due to an abiotic stimulus (e.g., drought, salinity or heat), enhanced trimethylation of K4 and acetylation of K9 occur at the histones, resulting in an open chromatin structure. In this state, RNA polymerase II (RNAPII) gains access to the DNA at the promotor region, enabling the transcription of abiotic stress response genes, including *PP2Cs* (protein phosphatase 2C), *Myb TFs* (Myb-related transcription factors), *CYPs* (cytochrome P450) and *Heat shock TF* (heat shock transcription factor).

## Data Availability

The data presented in this study are available in the Appendix A.

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
