# Peer review of "Drought-Stress-Related Reprogramming of Gene Expression in Barley Involves Differential Histone Modifications at ABA-Related Genes"

_ijms, 2023, doi:10.3390/ijms241512065_

Round 1

Reviewer 1 Report

Dear Authors

Present manuscript entitled “Drought stress-related reprogramming of gene expression in barley leaves involves loading of ABA-related genes with euchromatic marks H3K4me3 and H3K9ac” e investigate the drought response in barley plants (Hordeum vulgare, cv. Morex) at both epigenome and transcriptome levels in Leaves of barley plants. Comparison with differentially expressed genes enabled to identify those genes, which are loaded with the euchromatic marks and are induced in response to the drought treatment. The findings suggest that two members of the protein phosphatase 2C family (PP2Cs) are involved in the central regulatory machinery of ABA signaling. The present study is very well designed and results were presented nicely, although there are certain opportunities for further improvements, please find them below.

1.      In the Introduction- “Real time qPCR analyses with drought stressed rice seedlings showed an elevated expression of four histone acetyltransferases (HATs; OsHAC703, OsHAG703, OsHAF701, OsHAM701) and supporting Western-blot analysis revealed an enrichment of acetylation on H3K9, K18 and K27 as well as H4K5 in parallel to the increased OsHATs expression 27 . Loss of function studies of the Arabidopsis TRITHORAX-like factor ATX1 which trimethylates H3K4 leads to decreased levels of H3K4me3 at NCED3, a key enzyme in ABA biosynthesis, in drought stressed Arabidopsis plants, resulting in reduced ABA concentration 28. Furthermore, it could be shown, that during dehydration, NCED3 showed increased enrichment of nucleosomal H3K4me3 29. In two studies, Kim et al. (2008, 2012) showed enriched H3K4me3 and H3K9ac levels at the drought-inducible genes RD20 and RD29A where the enrichment levels correlated with the intensity of the stress (moderate vs severe) 18. Interestingly, during rehydration, the H3K9ac mark diminished fast and robustly while H3K4me3 decreased progressively 19 “ This information looks more suitable for the discussion section.

2.      Results and discussion is quite impressive, figures are nicely drawn and explained. Although conclusion need to be reframed. Conclusion should include the key findings only in a short text, which may be easier to understand for reader’s point of view. There is no need to include the references and discussion points.

3.      Section 4.1. Please include the information from where the seeds were procured.

4.      The experimental plants were grown in green house or in plant growth chamber, please include this information.

5.      How many pots were prepared and how many technical/biological replicates were considered.

Thank you

Author Response

Response to Reviewer 1

[General Comment] The present study is very well designed and results were presented nicely, although there are certain opportunities for further improvements, please find them below.

Response: Thank you very much. We read your suggestions carefully and tried to incorporate them in our manuscript.

[Comment 1] In the Introduction- “Real time qPCR analyses with drought stressed rice seedlings showed an elevated expression of four histone acetyltransferases (HATs; OsHAC703, OsHAG703, OsHAF701, OsHAM701) and supporting Western-blot analysis revealed an enrichment of acetylation on H3K9, K18 and K27 as well as H4K5 in parallel to the increased OsHATs expression 27 . Loss of function studies of the Arabidopsis TRITHORAX-like factor ATX1 which trimethylates H3K4 leads to decreased levels of H3K4me3 at NCED3, a key enzyme in ABA biosynthesis, in drought stressed Arabidopsis plants, resulting in reduced ABA concentration 28. Furthermore, it could be shown, that during dehydration, NCED3 showed increased enrichment of nucleosomal H3K4me3 29. In two studies, Kim et al. (2008, 2012) showed enriched H3K4me3 and H3K9ac levels at the drought-inducible genes RD20 and RD29A where the enrichment levels correlated with the intensity of the stress (moderate vs severe) 18. Interestingly, during rehydration, the H3K9ac mark diminished fast and robustly while H3K4me3 decreased progressively 19 “ This information looks more suitable for the discussion section.This information looks more suitable for the discussion section.

Response: As suggested, we took out this chapter and added this information to Discussion.

[Comment 2] Results and discussion is quite impressive, figures are nicely drawn and explained. Although conclusion need to be reframed. Conclusion should include the key findings only in a short text, which may be easier to understand for reader’s point of view. There is no need to include the references and discussion points.

Response: Thank you very much. We revised the conclusion as follows:  

“Barley has developed complex mechanisms to adapt to drought stress. One of these mechanisms includes histone modifications, particularly H3K9 acetylation, leading to the upregulation of specific genes. It is shown that during the early phase of drought stress, when chloroplasts remain active, a specific set of genes is loaded with H3K9ac and upregulated. Among these genes are ABA-related genes, such as those encoding PP2Cs involved in the central regulatory unit of ABA action. The results suggest that H3K9ac works in a more flexible manner in response to drought compared to K4 trimethylation in response to drought, indicating a potential role of H3K9 acetylation in the short stress memory of plants. Furthermore, the findings support the hypothesis that H3K9ac may act as a template for the trimethylation of K4.” [p.12]

[Comment 3] Section 4.1. Please include the information from where the seeds were procured.

Response: Thanks for your kind reminders. We added the information as follows:

“Barley (Hordeum vulgare cv. Morex) seeds, obtained from IPK Gatersleben (OT Gatersleben, Seeland, Germany),(…)” [p.13, Chapter 4.1]

[Comment 4] The experimental plants were grown in green house or in plant growth chamber, please include this information.

Response: Thanks for your kind reminders. We added the information as follows:

“Fifty Mitscherlich pots, each containing 12 seedlings and 1.5 kg ED73 soil (Einheitserdewerke Werkverband e.V., Germany), were placed in the greenhouse cabinet and grown under long day conditions (…)”[p.13, Chapter 4.1]

[Comment 5] How many pots were prepared and how many technical/biological replicates were considered.

Response: Thanks for your kind reminders. We added the information as follows:

“Fifty Mitscherlich pots, each containing 12 seedlings and 1.5 kg ED73 soil (Einheitserdewerke Werkverband e.V., Germany), were placed in the greenhouse cabinet and grown under long day conditions (…)”[p.13, 4.1]

“The experiment was performed eight times.”[p. 13, Chapter 4.1]

Reviewer 2 Report

General comments: 

  • I am not sure whether the method of citation used by the Authors is correct, but I leave the decision on this issue to the Editors. 
  Several specific comments: 

Due to the lack of line numbering in the downloaded pdf of the manuscript, my detailed comments have been included in the form of comments, directly in the file. 

Author Response

Response to Reviewer 2

[Comment on species names]  The reviewer pointed out several times the correct use of the abbreviations and the font style of the species names.

Response: Thank you for this kind reminder. We changed Hordeum vulgare to H. vulgare, Arabidopsis thaliana to A. thaliana, Brachypodium distachyon to B. distachyon, Maize to maize, Arabidopsis to Arabidopsis

[Comment 1]  I think the title in the presented form is intricate and too long, I suggest changing it.

Response: We changed it to: “Drought stress-related reprogramming of gene expression in barley involves differential histone modifications at ABA-related genes”

[Comment 2] This wording is not precise, I suggest either removing it or giving specific examples

Response: We changed it to: “Furthermore, the drought-related genes HvS40, HvA1, Hsp17 and P5CS2 were already induced compared to the well-watered controls.”

[Comment 3] lack of Keywords

Response: We added the Keywords:Keywords: ABA; drought stress; histone modifications; epigenome; transcriptome; Hordeum vulgare”

 [Comment 4] subscript

Response: Thank you for this kind reminder. We changed it.

[Comment 5] I suggest whole names in brackets and in order after abbreviations.

Response: Thank you for this kind reminder. We changed it.

[Comment 6] if there are the results it should be in the conclusion, here the purpose of the work only.

Response: We changed it to: “In this study, the genome-wide responses of the crop plant H. vulgare to drought stress at the levels of H3K4 trimethylation and H3K9 acetylation were investigated, showing epigenetic control of ABA-related stress responses.”

[Comment 7] This fragment fits more into the M&M chapter, I suggest moving it

Response: Thank you for this kind reminder. We changed it to:  “Barley (H. vulgare cv. Morex) plants, grown on soil, were exposed to drought by withholding water. This resulted in a continuous decrease in soil water content (Fig. 1A).

[Comment 8] this fragment fits more into the M&M chapter, I suggest moving it.

Response: Thank you for this kind reminder. We changed it to: “Drought stress results in a major reprogramming of gene expression 27. To map this process, expression of known marker genes of drought stress and leaf senescence was analyzed (Fig. 2).”

[Comment 9] Fig. 1B; Fig. 4A

Response: Thank you for this kind reminder. We changed it. Figure 1B to Fig. 1B and Figure 4A to Fig. 4A.

[Comment 10] not completed

Response: Thank you for this kind reminder. We changed it.

[Comment 11] This figure needs to be completed by the abbreviations explaining.

Response: We added information to the Figure 7: “Figure 7. Working model of the changes in histone modifications and chromatin structure during abiotic stress (created with BioRender.com). Shown are the histones modified with H3K4me3 (green circles with “Me3”) and H3K9ac (blue circles with “Ac”) in the nucleus. Due to an abiotic stimulus (e.g. drought, salinity or heat), enhanced trimethylation of K4 and acetylation of K9 occurs at the histones, resulting in an open chromatin structure. In this state, RNA polymerase II (RNAPII) gains access to the DNA at the promotor region, enabling the transcription of abiotic stress response genes, including PP2Cs (Protein Phosphatase 2C), MYB TFs (MYB transcription factors), CYP 450 (Cytochrome P450),  Heat shock TF (Heat shock transcription factor).”

[Comment 12] I propose: with slight modification

Response: We changed it to this expression.

 [Comment 13] How many plants were leaves taken from?

Response: “The relative chlorophyll content of 20 primary leaves from 20 plants was measured at the top, middle and bottom of the leave and the mean value was calculated.”

Reviewer 3 Report

The manuscript titled "Drought stress-related reprogramming of gene expression in barley leaves involves loading of ABA-related genes with euchromatic marks H3K4me3 and H3K9ac" aims to examine the genome-wide responses of the crop plant Hordeum vulgare to drought stress at the level of H3K4 trimethylation and H3K9 acetylation. The study concluded that Barley's adaptive response to drought stress involves complex histone modifications, particularly H3K9 acetylation, which influences gene up-regulation, especially those related to ABA signaling. The dynamic nature of H3K9ac indicates its potential involvement in short-term stress memory in plants, potentially acting as a scaffold for K4 trimethylation. While the topic is of significant relevance and general interest to the journal's readership, several concerns need to be addressed before publication.

·         The authors are highly recommended to avoid using a personal pronoun (e.g., We, our, etc.); they can use the third party in the past tense's passive voice.

·         The authors are strongly advised to carefully review the manuscript to address grammar and other editing issues.

·         Avoid short paragraph in the introduction.

·         In the Material and Methods section, it is important to include or complete the sources of all chemicals, software, and equipment by adding the city, state, and country information. This additional detail provides readers with specific information about where these items were sourced, ensuring transparency, and facilitating reproducibility.

·         In Figure 2, the legend is not clear both are black.

·         Citation in the text must follow the journal style.

Minor editing of English language required

Author Response

Response to Reviewer 3

[General Comment] While the topic is of significant relevance and general interest to the journal's readership, several concerns need to be addressed before publication.

Response: Thank you very much. In the following, we will try to consider your helpful and constructive comments.

[Comment 1]  The authors are highly recommended to avoid using a personal pronoun (e.g., We, our, etc.); they can use the third party in the past tense's passive voice.

Response: Thank you for this kind reminder. We changed all the personal pronouns to the third party.

[Comment 2] The authors are strongly advised to carefully review the manuscript to address grammar and other editing issues.

Response: We went through the entire manuscript and eliminated grammatical mistakes and editing issues.   

 [Comment 3] Avoid short paragraph in the introduction.

Response: We eliminated the short paragraphs in the introduction.

 [Comment 4]  In the Material and Methods section, it is important to include or complete the sources of all chemicals, software, and equipment by adding the city, state, and country information. This additional detail provides readers with specific information about where these items were sourced, ensuring transparency, and facilitating reproducibility.

Response: Thanks for your kind reminders. We added the missing information as follows:

“Barley (Hordeum vulgare cv. Morex) seeds, obtained from IPK Gatersleben (OT Gatersleben, Seeland, Germany), (…)” [p.13, Chapter 4.1]

„(…) 1.5 kg ED73 soil (Einheitserdewerke Werkverband e.V., Sinntal-Altengronau, Germany) (…)“ [p.13, Chapter 4.1]

“In general, all DNA and RNA concentrations were measured with the Nanospectrophotometer (NanoPhotometer® NP80, Implen, Munich, Germany)” [p.14, Chapter 4.3]

“(…) using the Covaris M220 Focused Ultrasonicator (Covaris LLC, Woburn, Massachusetts, USA) with following settings (…)” [p.14, Chapter 4.4]

“(…) with the KAPA SYBR fast qPCR Mastermix (KAPA Biosystems, Inc., Wilmington, Massachusetts) (…)” [p. 14, Chapter.4.3]

The bioinformatics software tools were cited.

[Comment 5]  In Figure 2, the legend is not clear both are black.

Response: Thanks for your kind reminders. We corrected it.

[Comment 6]  Citation in the text must follow the journal style.

Response: Thanks for your kind reminders. We corrected it.

[Comment 7]  Minor editing of English language required

Response: We carefully reread the manuscript and eliminated grammatical and language mistakes.